# GRAPH-RELATIONAL FEDERATED LEARNING: ENHANCED PERSONALIZATION AND ROBUSTNESS

## ABSTRACT

Hypernetwork has recently emerged as a promising technique to generate personalized models in federated learning (FL). However, existing works tend to treat each client equally and independently — each client contributes equally to learning the hypernetwork, and their representations are independent in the hypernetwork. Such an independent treatment ignores topological structures among different clients, which are usually reflected in the heterogeneity of client data distribution. In this work, we propose *Panacea*, a novel FL framework that can incorporate client relations as a graph to facilitate learning and personalization using graph hypernetwork. Empirically, we show *Panacea* achieves state-of-the-art performance in terms of both accuracy and speed on multiple benchmarks. Further, *Panacea* improves the robustness by leveraging the client relation graph. Specifically, it (1) generalizes better to the novel clients outside of the training and (2) is more resilient to label-flipping attacks, which is also proved by our theoretical analysis.

## 1 INTRODUCTION

With the rise of the Internet of Things (IoT), a massive amount of data is being collected from geographically distributed sources, including mobile phones, wearable sensors, and other IoT devices. This data is highly informative and can be used for various AI-based applications, including predicting health events such as the risk of heart attacks AbdulRahman et al. (2020). To optimize these devices' storage and computational capabilities, storing data locally and pushing more computation to IoT devices is preferred. Additionally, privacy-preserving AI is imperative to comply with data privacy regulations like GDPR Voigt & Von dem Bussche (2017). Federated Learning (FL) has emerged as a popular paradigm for training statistical models over distributed devices/clients while keeping the training data local Kairouz et al. (2021).

In typical federated learning, each client/device holds a dataset for local training, and a server aggregates gradients from clients for global model updates McMahan et al. (2017). A unique global model can then be applied to all clients McMahan et al. (2017); Mothukuri et al. (2021). However, this paradigm might be sub-optimal in practice, as the clients' data distributions are usually heterogeneous. To address this issue, personalized federated learning (PFL) is proposed to train a personalized local model for each client while each client can still leverage the knowledge from other clients in the federation Tan et al. (2022). However, there is a key challenge in PFL: *How to enable beneficial collaborative training while preserving the uniqueness of the clients yet achieving communication efficiency.*

A promising approach to balance information sharing and uniqueness preservation among clients is Personalized Federated Hypernetworks (pFedHN) Shamsian et al. (2021). Hypernetworks Ha et al. (2017) are neural networks that generate model parameters for another set of deep neural networks. In pFedHN, a multi-layer perceptron (MLP) hypernetwork takes the local client's representation as input and generates personalized model weights corresponding to each client. By providing a mapping from the clients' embedding space to the clients' model parameter space, pFedHN achieves a better generalization for clients that were unseen during training.

Despite providing a principled approach to personalized model learning, pFedHN has limitations. It treats every client equally and independently, ignoring the potential relations of data heterogeneity across the clients. Specifically, each client's representation and corresponding local gradients are

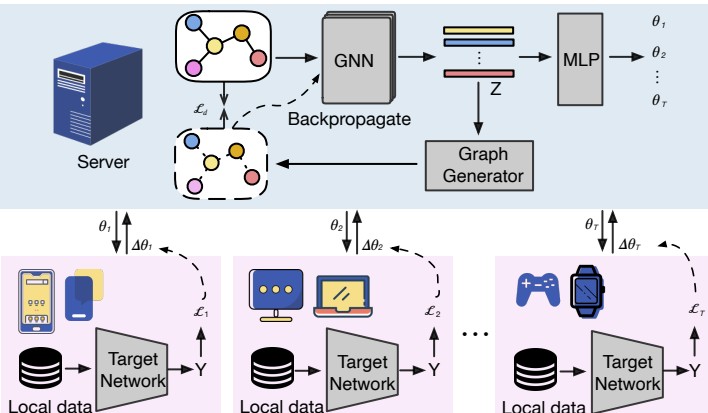

Figure 1: Illustration of *Panacea*. The framework has two components 1) a graph hypernetwork consisting of a GNN encoder that embeds clients based on their relationships, and an MLP that generates local model weights based on the embeddings; 2) a graph generator that reconstructs the client relation graph to preserve the clients' uniqueness. Local models and hypernetwork parameters are jointly optimized using alternating optimization.

regarded as independent data samples for learning the hypernetwork. However, the clients' data distributions are often interdependent, which often forms a graph where each client is a node, and the links reflect the data distribution dependency between clients. More importantly, such a graph is naturally accessible in real-world applications with IoT networks. For example, in road network traffic forecasting in the Bay Area, sensors measuring traffic speed can be modeled as nodes, with physical proximity between two sensors as a link Li et al. (2018). By modeling the clients' relations as a graph, achieving more effective knowledge sharing among the clients in the federation is possible.

This paper presents *Panacea*, a novel framework for personalized federated learning based on hypernetworks. The framework comprises a graph neural network (GNN) module and a graph generator. The GNN module takes the client representation and their relationships as input to generate local model weights. To preserve the "uniqueness" of different clients, we introduce a graph generator that distinguishes discrepancies among clients in the federation. The graph generator uses the encoding of the local model weights, i.e., client embeddings, to reconstruct the client relation graph. With a GNN module and a graph generator, *Panacea* enables (1) effective knowledge sharing among clients, even when two clients have low similarities, i.e., distant neighbors in the relation graph, (2) avoids negative influence between clients with significant distribution discrepancies while ensuring personalization performance.

In summary, this paper makes the following contributions. First, we propose a novel personalized federated learning framework, called *Panacea*, that leverages the inherent relationships among clients to encourage effective knowledge sharing while preserving each client's uniqueness. Second, unlike other methods that incur extra communication costs, *Panacea* requires no additional communication due to the introduced graph hypernetwork. Third, we conduct extensive empirical studies on many synthetic and real-world datasets across various learning tasks, demonstrating that *Panacea* outperforms prior state-of-the-art federated learning algorithms. Furthermore, we show that *Panacea* improves the robustness of the system in two ways: (1) better generalization to novel clients outside of the training and (2) more resilience to malicious clients, as demonstrated by both experiments and theoretical analysis.

## 2 RELATED WORK

**Federated learning with hypernetworks.** The most related prior work is pFedHN Shamsian et al. (2021) which uses hypernetworks to generate personalized model weights. The major difference between our work and pFedHN is that we further incorporate client interdependency. We will show that leveraging the client relation graph, *Panacea* can boost the performance of PFL from various aspects, including better generalizability to unseen clients and robustness to malicious clients using label-flipping attacks – a representative attacks in federated learning settings Fung et al. (2020).

**Federated learning with graphs.** Several prior works do federated learning with graphs. Most of them focus on graph learning problems on graph-structured data Meng et al. (2021); He et al. (2021). For example, FedGraphNN He et al. (2021) provides an open-source FL system for GNNs, which enables federated training over various GNNs. Meng *et al.* Meng et al. (2021) proposes a GNN-based federated learning architecture that attempts to capture complex spatial-temporal data dependencies among multiple participants. The closest to us is SFL Chen et al. (2022), which also attempts to leverage graphs to boost the performance of personalized federated learning. SFL naively updates the client weights by averaging over its neighbors. In contrast, we generalize hypernetwork-based approach with a graph learning module equipped with a generator, which can incorporate the relational graph of clients for generating local model weights. Following the standard FL setting, we do not require all clients to participate in each round. More importantly, with a dedicated graph reconstruction component, our approach exhibits better performance, particularly robustness in adversarial settings.

# 3 PROPOSED FRAMEWORK: PANACEA

## 3.1 PROBLEM FORMULATION AND NOTATIONS

Our objective is to train personalized models collaboratively for a set of $\mathcal{T}$ clients, each with its unique local dataset. Each client $t \in \mathcal{T}$ has a distribution $\mathcal{P}_t$ on $\mathcal{X}_t \times \mathcal{Y}_t$. Since the local data distributions $\mathcal{P}_t \in \mathcal{T}$ differ, it's natural to fit a single model to each data distribution. We assume that each client has access to $|\mathcal{D}_t|$ IID data points drawn from $\mathcal{P}_t$, and client $t$'s local dataset can be represented as $\mathcal{D}_t = \{(\mathbf{x}_j^{(t)}, y_j^{(t)})\}_{j=1}^{|\mathcal{D}_t|}$. The relationship among clients/devices is described by a graph with the adjacency matrix $\mathbf{A} = [\mathbf{A}_{uv}]$, where $u$ and $v$ index the clients in the graph. Let $\ell_t : \mathcal{Y} \times \mathcal{Y} \to \mathbb{R}_+$ describe the loss function corresponding to client $t$, and $\mathcal{L}_t(\theta_t) = \frac{1}{|\mathcal{D}_t|} \sum_j \ell_t(\mathbf{x}_j, y_j; \theta_t)$ is client $t$'s average loss over its personal training data, where $\theta_t$ denotes the model weights of client $t$. The loss function varies for different tasks, e.g., *cross-entropy* is commonly used for classification while *mean square error* is preferred for forecasting. The goal is to solve the following optimization problem:

$$\Theta^* \in \arg\min_{\Theta} \frac{1}{|\mathcal{T}|} \sum_{t=1}^{|\mathcal{T}|} \mathbb{E}_{(\mathbf{x},y)\sim\mathcal{P}_t} \left[\ell_t(\mathbf{x}_j, y_j; \theta_t)\right]. \tag{1}$$

During training, the optimization is carried out on finite training samples as follows:

$$\arg\min_{\Theta} \frac{1}{|\mathcal{T}|} \sum_{t=1}^{|\mathcal{T}|} \mathcal{L}_t(\theta_t) := \frac{1}{|\mathcal{T}|} \sum_{t=1}^{|\mathcal{T}|} \frac{1}{|\mathcal{D}_t|} \sum_{j=1}^{|\mathcal{D}_t|} \ell_t(\mathbf{x}_j, y_j; \theta_t), \tag{2}$$

where $\Theta = \{\theta_t\}_{t=1}^{\mathcal{T}}$ are the set of personalized model parameters for all clients.

**Remark.** In this paper, we assume the existence and accessibility of a client relation graph that reflects the similarity of local models. we argue such an assumption is practical in many real-world scenarios. For example, in news recommender systems Liu et al. (2010), clients are mobile phones containing user data, which can be connected as a graph via social networks. Intuitively, neighbors on the social network will have similar usage patterns of phones. For the cases where the graph naturally exists but is unobserved, we leave it to future work that might incorporate relation inference models Peng et al. (2020) into our framework to estimate the graph during federated learning.

## 3.2 OUR FRAMEWORK

**Pipeline overview.** Our *Panacea* framework is composed of a GNN encoder, an MLP, and a graph generator, as shown in Fig. 1. The GNN encoder, denoted as $E(\cdot; \psi_e)$, takes the client's initial embedding and the client relationship (i.e., the adjacency matrix $\mathbf{A}$) as input. It generates informative client embeddings $\mathbf{Z}$ after multiple GNN propagation steps. The MLP, $M(\cdot; \psi_m)$, generates local model weights $\theta_t$ for each client $t$ based on the client embeddings. Finally, a graph generator $D(\cdot)$ preserves the local preferences of each client by reconstructing the graph relation of the clients.

**Training objective for the server.** In *Panacea*, the GNN encoder and the MLP together is considered as the graph hypernetwork denoted as $\mathcal{GH}(\cdot; \psi) = M(\cdot; \psi_m) \circ E(\cdot; \psi_e)$, where $\psi = \{\psi_e, \psi_m\}$.

To learn the hypernetwork, *Panacea* optimize the following objective:

$$\min_{\psi_e, \psi_m} \mathcal{L}_{\mathcal{GH}}(E, M) + \lambda_d \mathcal{L}_d(D, E), \tag{3}$$

where $\mathcal{L}_{\mathcal{GH}}(E, M)$ is the graph hypernetwork loss and $\mathcal{L}_d(D, E)$ is the graph reconstruction loss, and $\lambda_d$ is the weight to balance these two terms. Note that the graph generator $D$ is only involved during training to encourage the hypernetwork to preserve clients' uniqueness.

**Graph hypernetwork.** We use $h_t^{(L)}$ to denote the embedding of client $t$ at the $L$'s layer of the GNN encoder. Given client initial embedding $h_t^{(0)}$ and the graph relation of the clients $\mathbf{A}$, the graph hypernetwork outputs the local model weights for client $t$ via $\mathcal{GH}(h_t^{(0)}, \mathbf{A}; \psi)$. Collaboratively, the graph hypernetwork learns a family of personalized local models for clients $\{\mathcal{GH}(h_t^{(0)}, \mathbf{A}; \psi) | t \in \mathcal{T}\}$. To train it, we use the mean squared error between the hypernetwork predicted model parameters and the local model parameters gathered from the clients' local updates in this communication round which denoted as $\theta_t$. Formally, the graph hypernetwork loss is the following,

$$\mathcal{L}_{\mathcal{GH}} = \frac{1}{2|\mathcal{T}|} \sum_{t=1}^{|\mathcal{T}|} ||\theta_t - \mathcal{GH}(h_t^{(0)}, \mathbf{A}; \psi_e, \psi_m)||_2^2. \tag{4}$$

**Graph generator.** We train the graph generator using a graph reconstruction loss. Specifically, we realize it with an inner-product operator, which admits the client embeddings as the input and reconstructs the graph relation of clients. The intuition is that good client embeddings should be able to preserve the "affinity" between clients. Therefore, they can inform us of the graph relation of clients. We independently sample client indices $u$ and $v$ from the marginal client index distribution $p(c)$ and compute the reconstruction loss as,

$$\mathcal{L}_d = \mathbb{E}_{u,v \sim p(c)}[-\mathbf{A}_{uv} \log \sigma(\hat{\mathbf{z}}_u^\mathsf{T}\hat{\mathbf{z}}_v) - (1 - \mathbf{A}_{uv}) \log(1 - \sigma(\hat{\mathbf{z}}_u^\mathsf{T}\hat{\mathbf{z}}_v))], \tag{5}$$

where $\mathbf{z}_u = E(h_u; \psi_e)$ is the embedding of client $u$, $\hat{\mathbf{z}}_u$ is the transformation of $\mathbf{z}_u$ with the MLP and $\sigma(x) = \frac{1}{1+\exp^{-x}}$ is the sigmoid function.

**Training objective for the clients.** Each client $i$ is aim to reduce its own local task. Under the constrain that the local model's parameters are generated by the hypernetwork, we can formulate the learning objective of all clients as the following,

$$\arg\min_{\Theta} \frac{1}{|\mathcal{T}|} \sum_{t=1}^{|\mathcal{T}|} \mathcal{L}_t(\theta_t) = \arg\min_{\Theta = \{\mathcal{GH}(h_t, \mathbf{A}; \psi_e, \psi_m)\}_t} \frac{1}{|\mathcal{T}|} \sum_{t=1}^{|\mathcal{T}|} \frac{1}{|\mathcal{D}_t|} \sum_{j=1}^{|\mathcal{D}_t|} \ell_t(\mathbf{x}_j, y_j; \theta_t). \tag{6}$$

Essentially, the goal of personalized federated learning is to enable each client to benefit from knowledge available from other clients to get a better approximation for their local model parameters $\theta_t$ — each client in the federation obtains a solitary model that can have a better generalization ability to unseen samples.

**Joint optimization of $\Theta$ and $\psi$.** By sharing the graph hypernetwork parameters, *Panacea* can enforce effective knowledge sharing among the clients in the federation. However, jointly optimizing formula (3) and formula (6) can be difficult due to the non-convexity of each term. Specifically, both graph hypernetwork and local models are deep neural networks. Thus optimizing them are non-convex problems. In addition, there is no explicit and stable supervision signal for learning the graph hypernetwork parameters $\psi$, as $\Theta$ is dynamically changing in each communication round before convergence. To facilitate the joint optimization of (3) and (6), we leverage following observations:

*Observation 1:* With a fixed hypernetwork $\psi$, the updates over $\Theta$ only depend on the local dataset $\{\mathcal{D}_t\}_{t \in \mathcal{T}}$ as illustrated in formula (6).

*Observation 2:* The gradient of the hypernetwork $\psi$ only depends on $\Theta$ rather than on the client dataset $\{\mathcal{D}_t\}_{t=1}^{\mathcal{T}}$, as shown in formula (3), (4), and (5).

The above observations suggests that it is natural to use alternating optimization approach Smith et al. (2017) to jointly solve formula (3) and formula (6) for learning the personalized models and graph hypernetwork parameters. Specifically, using the chain rule, we can have the gradient of $\psi_e$ and $\psi_m$ from formula (3), (4), and (5):

$$\nabla_{\psi_e}\{\mathcal{L}_{\mathcal{GH}} + \lambda_d \mathcal{L}_d\} = \nabla_{\psi_e}\mathcal{L}_{\mathcal{GH}} + \lambda_d \nabla_{\psi_e}\mathcal{L}_d = (\nabla_{\psi_e}\theta_t)^\mathsf{T}\nabla_{\theta_t}\mathcal{L}_{\mathcal{GH}} + \lambda_d \nabla_{\psi_e}\mathcal{L}_d; \tag{7}$$

$$\nabla_{\psi_m} \{\mathcal{L}_{\mathcal{GH}} + \lambda_d \mathcal{L}_d\} = \nabla_{\psi_m} \mathcal{L}_{\mathcal{GH}} = (\nabla_{\psi_m} \theta_t)^\mathsf{T} \nabla_{\theta_t} \mathcal{L}_{\mathcal{GH}}. \tag{8}$$

From formula (7), we can observe that only the first term of gradient $\psi_e$ needs the local updates from the client. Therefore, we can simply use a general update rule $(\nabla_{\psi_e} \theta_t^\mathsf{T}) \triangle \theta_t$ to approximate the first term of $\psi_e$, i.e., $\triangle \psi_e = (\nabla_{\psi_e} \theta_t^\mathsf{T}) \triangle \theta_t$, where $\triangle \theta_t$ denotes the change of client $t$'s model parameters after a round of local updates. For the second term $\mathcal{L}_{d_t}$, it does not need the gradients from the client; therefore, easily computable on the server side. Similarly, the gradient update of $\psi_m$ can be approximated based on the update rule: $\triangle \psi_m = (\nabla_{\psi_m} \theta_t^\mathsf{T}) \triangle \theta_t$. Extensive literature has shown the benefits of performing multiple local optimization steps per communication round in terms of both convergence rate and final accuracy McMahan et al. (2017); Shamsian et al. (2021). Thus, in *Panacea*, we perform multiple local updates. $K_c$ and $K_s$ steps for the client models and the hypernetowrk at server.

Algorithm 1 demonstrates the detailed procedure of our proposed framework for learning the graph hypernetworks and the personalized local models for the clients. In each communication round, the clients download the latest

---

**Algorithm 1** Panacea

**Require:** $\alpha$ — learning rates, $\eta$ — client learning rate, $R$ — number of rounds, $K_c$, $K_s$ — number of local rounds for clients and server.
**Ensure:** Personalized models $\{\theta_1, \theta_2, \cdots, \theta_{\mathcal{T}}\}$, and a graph hypernetwork model $\psi$.
1: **for** each communication round $i \in [R]$ **do**
2:      sample a subset of clients $S_t \subset [\mathcal{T}]$
3:      **for** each client $t \in S_t$ **do**
4:          $\theta_t = \mathcal{GH}(h_t^{(0)}, \mathbf{A}; \psi)$, and $\tilde{\theta}_t = \theta_t$
5:          **for** each local step $k \in [K_c]$ **do**
6:              sample a mini-batch $B \subset \mathcal{D}_t$
7:              $\tilde{\theta}_t = \tilde{\theta}_t - \eta \nabla_{\tilde{\theta}_t} \mathcal{L}_t(B)$
8:          **end for**
9:          $\triangle \theta_t = \tilde{\theta}_t - \theta_t$
10:      **end for**
11:      **for** each local step $k \in [K_s]$ **do**
12:          $\psi_e = \psi_e - \alpha \nabla_{\psi_e} \theta_t^\mathsf{T} \triangle \theta_t - \alpha \lambda_d \nabla_{\psi_e} \mathcal{L}_d$
13:          $\psi_m = \psi_m - \alpha \nabla_{\psi_m} \theta_t^\mathsf{T} \triangle \theta_t$
14:      **end for**
15: **end for**
16: Return the personalized models $\{\tilde{\theta}_t\}_{t \in \mathcal{T}}$, and the graph hypernetwork model $\psi$.

---

personalized models from the server, then use local SGD to train $K_c$ local steps to update the local model weights. After that, each client will upload their model updates to the server. Accordingly, the server will train $K_s$ local steps to update the hypernetwork parameters. Note that, similar to pFedHN, our Panacea incurs no additional communication cost compared with traditional FL methods, like FedAvg McMahan et al. (2017) (detailed analysis is in Appendix B).

### 3.3 THEORETICAL ANALYSIS

In this section, we analyze the linear case of Panacea and compare it with the pFedHN to emphasize the benefit of introducing the client relation graph. All proofs are delayed into Appendix A.

**Notations.** We have $n$ clients. For each client $i$, data $\mathbf{x}_i \in \mathbb{R}^d$ with a dimension $d$ follows a standard Gaussian distribution $x_i \sim \mathcal{N}(0, I_d)$ and labels $y_i$ are generated by a ground truth linear model with a parameter $\theta_i^*$, i.e., $y_i = \mathbf{x}_i^\top \theta_i^*$. We use $\Theta^* = [\theta_1^*, \cdots, \theta_n^*]$ to denote all ground truth parameters. For a model with parameter $\Theta = [\theta_1, \cdots, \theta_n]$, its expected risk at client $i$ is $\mathcal{R}(\theta_i) := \mathbb{E}_{(x_i, y_i)}(y_i - x_i^\top \theta_i)^2 = \|\theta_i^* - \theta_i\|_F^2$. The averaged expected risk across clients is $\mathcal{R}(\Theta) := \frac{1}{n} \mathcal{R}(\theta_i) = \frac{1}{n} \|\Theta^* - \Theta\|_F^2$. pFedHN generate models via a linear hypernetwork with a latent dimension $k$ and parameters $W \in \mathbb{R}^{d \times k}$ and $V := [v_1, \cdots, v_n] \in \mathbb{R}^{k \times n}$ which denote the weight of hypernetwork and the client embeddings respectively. The client parameters are generated by applying the hypernetwork to each client embedding, i.e., $\Theta = WV$. It is typically assumed that $d > n > k$. Our *Panacea* uses a GNN as the hypernetwork leading to the following decomposition of $\Theta$ as $\Theta = W_L(\cdots W_2(W_1(V\tilde{G}))\tilde{G} \cdots \tilde{G})$ where $L$ is the number of GNN layers and in $l$th layer, $W_l$ is per node transformation and $\tilde{G}$ is aggregation operation. For simplicity, we use mean aggregation, i.e., $[\tilde{G}]_{i,j} = \frac{1}{N(i)+1} 1_{[j \in N(i) \cup \{i\}]}$ where $N(i) := \{j | \mathbf{A}_{ij} = 1\}$ is neighborhood of node $i$ in the graph. We can express $\Theta$ in a compact form via $\Theta = WVG$ where $W = W_L \cdots W_1$ and $G = \tilde{G}^L$.

**Optimum.** As observed in the original paper of pFedHN, learning a linear hypernetwork by minimizing the expected risk $\mathcal{R}(\Theta) \propto \|\Theta^* - \Theta\|_F^2$ with $\Theta = WV$ is equivalently to solving the rank $k$ approximation of the matrix $\Theta^*$. Thus the optimum is achieved when it learns the top $k$ PCA components, i.e., $\Theta_k^* = P\mathbf{diag}(\lambda_1, \cdots, \lambda_k, 0, \cdots, 0)Q^\top$ where $P\mathbf{diag}(\lambda_1, \cdots, \lambda_n)Q^\top$ is the singular

value decomposing (SVD) of $\Theta^*$. Theorem 3.1 shows that *panacea*, although using a GNN, has the same optimum as long as the graph is not generated, i.e., $\mathbf{rank}(\tilde{G}) = n$.

**Theorem 3.1.** *Optimization* $\min_{\Theta = WVG} \mathcal{R}(\Theta)$ *has an unique minimum* $\Theta_k^*$, *if* $\mathbf{rank}(\tilde{G}) = n$.

**Robustness.** In the following, we show leveraging extra knowledge of the client relations can bring us a gain of robustness. To facilitate the analysis, we will assume the ground truth model parameters can be decomposed via $\Theta^* = W^* V^*$ where $W^* \in \mathbb{R}^{d \times k}$ is the ground truth hypernetwork with elements i.i.d from $\mathcal{N}(0, 1)$ and $V^* \in \mathbb{R}^{k \times n}$ is the ground truth client embeddings satisfies (1) *Consistency*: be consistent with the graph structure as $V^* G \simeq V^*$ and (2) *Gaussianity*: each entry has a Gaussian marginal $\mathcal{N}(0, 1)$. For the algorithms, we assume the initialization of the model hypernetwork, $W_0$ and client embeddings, $V_0$ both have i.i.d Gaussian entries $\mathcal{N}(0, 1)$. For the attack setting, label flip attack is originally defined in the context of classification and gives zero gradient in expectation since the label is randomly flipped. For simplicity, we assume the attacked client provides no gradient during training. Further, say $m$ clients (client 1 to $m$) are attacked. We call $\alpha := \frac{m}{n} < 1$, *the attack ratio*. Further, we restrict our analysis in a not too strong attack where $n - m \geq k$ implying the possibility of recovering the optimal hypernetwork with the remaining unattacked clients. Thus both pFedHN and Panacae will learns $W^*$ (under an equivalence of orthogonal transform). So we will only analyze to optimization of clients' embeddings $V$. Optimization with the unattacked clients is equivalent to minimize the loss $\mathcal{R}_{\texttt{uatk}}(\Theta) = \frac{1}{n} \|(\Theta^* - \Theta)_{(m+1):n}\|_F^2$ where we use the subscripts $(m+1) : n$ to denote $m+1$'th to $n$'th columns of the matrix. Let $\hat{\Theta}_{\texttt{mlp}}$ and $\hat{\Theta}_{\texttt{gnn}}$ be the learned model via pFedHN and Panacea, i.e., $\hat{\Theta}_{\texttt{mlp}} \in \operatorname{argmin}_{\Theta = W^* V} \mathcal{R}_{\texttt{uatk}}(\Theta)$ and $\hat{\Theta}_{\texttt{gnn}} \in \operatorname{argmin}_{\Theta = W^* VG} \mathcal{R}_{\texttt{uatk}}(\Theta)$.

**Theorem 3.2.** *For pFedHN, in expectation, its risk is* $\mathbb{E}[\mathcal{R}(\hat{\Theta}_{\texttt{mlp}})] = 2dk\alpha$.

**Theorem 3.3.** *For Panacea,* $\mathbb{E}[\mathcal{R}(\hat{\Theta}_{\texttt{gnn}})] = \frac{2dk}{n} \|(I - G_{(m+1):n} G_{(m+1):n}^\dagger) G_{1:m}\|_F^2$.

**Lemma 3.1.** *Denote* $r(G, m) := \|(I - G_{(m+1):n} G_{(m+1):n}^\dagger) G_{1:m}\|_F^2$ *which is the distance of $G$'s first $m$ columns to the span of the rest columns of $G$. We have $0 \leq r(G, m) \leq m$. Further, the upperbound is achieved, iff $G_{1:m} = I_{1:m}$ and $G_{1:m}^\top G_{(m+1):n} = \mathbf{0}$. The lowerbound is achieved iff* $\mathbf{col}(G_{1:m}) \subset \mathbf{col}(G_{(m+1):n})$.

Based on the theorems, we make a few remarks: ❶ Panacea has a smaller expected risk than pFedHN, i.e., $\mathbb{E}[\mathcal{R}(\hat{\Theta}_{\texttt{gnn}})] \leq \mathbb{E}[\mathcal{R}(\hat{\Theta}_{\texttt{mlp}})]$, due to the fact that $r(G, m) \leq m$ from lemma 3.1. ❷ pFedHN's loss is linearly scale with the attack ratio $\alpha$ while Panacea does not. ❸ Instead, Panacea's risk is related to the graph structure. We discuss two extreme cases: (1) *Empty graph* with no edges which is totally non-informative. Then $G$ becomes the identity $I$ which leads to Panacea degenerate to pFedHN with $\mathbb{E}[\mathcal{R}(\hat{\Theta}_{\texttt{gnn}})] = \mathbb{E}[\mathcal{R}(\hat{\Theta}_{\texttt{mlp}})]$; (2) *Clique graph* with all nodes are connected, which is highly informative, since knowing one client's embedding equals to knowing all of them. Thus we can always fully recover the client embeddings from the unattacked ones. Mathematically, $G = \frac{1}{n} \mathbf{1}\mathbf{1}^\top$ which leads $r(G, m) = 0$ for any $m < n$ implying $\mathbb{E}[\mathcal{R}(\hat{\Theta}_{\texttt{gnn}})] = 0$ from Theorem 3.3.

## 4 EMPIRICAL STUDIES

We evaluate *Panacea* and all baseline methods across various tasks from different application domains.

**1) Synthetic data for classification.** *FL* is a synthetic binary classification for federated learning. We borrow the synthetic data distribution from a prior domain adaptation work Xu et al. (2021). For each client $t$, a 2-dimensional unit vector $[a_t, b_t]$ was randomly generated as the client embedding, and the angle of the unit vector was represented as $\omega_t = \arcsin(\frac{b_t}{a_t})$. Positive samples $(\mathbf{x}, 1, t)$ and negative samples $(\mathbf{x}, 0, t)$ are sampled from two different 2-dimensional Gaussian distributions, $\mathcal{N}(\mu_{t,1}, \mathbf{I})$ and $\mathcal{N}(\mu_{t,0}, \mathbf{I})$, respectively, where $\mu_{t,1} = [\frac{\omega_t}{\pi} a_t, \frac{\omega_t}{\pi} b_t]$ and $\mu_{t,0} = [-\frac{\omega_t}{\pi} a_t, -\frac{\omega_t}{\pi} b_t]$. Then we construct the client relation graph with a Bernoulli distribution, i.e., $\mathbf{A}_{uv} \sim \mathbf{Bern}(0.5 a_u a_v + 0.5 b_u b_v + 0.5)$. The generation process ensures that the datasets

of all clients are non-IID and the adjacent clients in the graph have similar decision boundaries for classification.

**2) Car image data for classification.** *Comprehensive Cars* (CompCars) Yang et al. (2015) contains 136,726 images of cars with labels including 4 car types (MPV, SUV, sedan, and hatchback), 5 viewpoints (front (F), rear (R), side (S), front-side (FS), and rear-side (RS)), and years of manufacture (YOMs, ranging from 2009 to 2014). We follow the data splitting from Xu et al. (2021), and each client has car images only from one viewpoint and one YOM. The task is to predict the car type based on the image. Two clients are connected if either their viewpoints or YOMs are identical/nearby. For example, client A and B are connected if A's YOM is 2009, and B's is 2010.

**3) State network temperature data for regression.** *TPT-48* is a real-world dataset for temperature prediction. It contains the monthly average temperature over 48 contiguous states in the US from 2008 to 2019. We use the data processed by Washington Post WP [1]. The task is to forecast the next 6 months' temperature given the previous first six months' temperature. Due to the diverse geographical environments, the collected temperature datasets from various states are inherent non-IID. The geographically adjacent states form links in the graph.

**4) Road network traffic data for forecasting.** *PEMS-BAY* and *METR-LA* are two real-world datasets for traffic forecasting Li et al. (2018); Xu et al. (2021). The task is to predict the traffic speed in the following 12 steps of each sequence given the first 12 steps. Each sensor/loop-detector realizes the node for these two traffic datasets, and the adjacent sensors/detectors form links in the graph. In particular, the relational graph was constructed based on the distance of the clients Meng et al. (2021). More dataset details are in Appendix C.

### 4.1 EXPERIMENT SETTINGS

**Baselines.** We compare *Panacea* against various personalized federated learning (PFL) approaches. Our baselines include *Hypernetwork-based* methods: **pFedHN** Shamsian et al. (2021), which treats each client equally and independently and realizes the hypernetwork with a MLP. *Graph-based* methods: **SFL** Chen et al. (2022) that enhance personalization by using graph relation and doing information aggregation from nearby clients. *Standard* FL methods: 1) **FedAvg** McMahan et al. (2017), the classical federated learning algorithm based on federated averaging; 2) **Per-FedAvg** Fallah et al. (2020b) that personalizes client models via meta-learning; 3) **pFedMe** T Dinh et al. (2020) that improves personalization by adding a regularization term in the objective function; **FedAvg-finetune** Collins et al. (2021) that realizes personalization by incorporating fine-tuning over federated averaging.

**Implementations.** Our framework consists of three modules: a GNN encoder, an MLP, and a graph generator (see Figure 1 in Sec. 3). For evaluation, we use a 3-layer GNN encoder with hidden dimension 100 and a 3-layer MLP for generating local model parameters. The graph generator is instantiated with an inner-product operator. To illustrate the necessity of incorporating the graph generator module, we evaluate a variant of our framework with the GNN encoder and the MLP module only, called *Panacea*-GN. The client embedding dimension was fixed to 100 for all datasets. For the baselines, we use the public original implementations. More details can be found in Appendix D.

**Evaluation metrics.** The evaluation for *Panacea* is geared towards personalized model performance, generalization capability to unseen clients, and robustness to malicious clients under label-flipping attacks. The evaluating metric of predictive performance varies in different learning tasks. We evaluate the performance with the classification accuracy of the held-out test set averaged over all clients for classification tasks and the Mean Square Error (MSE) for regression and forecasting tasks, respectively. To evaluate the generalization capability to unseen clients, we hold out 20% of the clients as novel clients and evaluate corresponding model performance, i.e., classification accuracy or MSE. We introduce malicious clients under various attack ratios during training to assess robustness. Specifically, the malicious clients randomly flip the labels of their local data samples for training. In each round, they report the jamming local-updates to the server, attempting to disrupt the entire federated training. The attack ratio ranges from 0% to 50%. All the reported results, e.g.,

---

[1] The raw data is from the National Oceanic and Atmospheric Administration's Climate Divisional Database (nClimDiv) and Gridded 5KM GHCN-Daily Temperature, and Precipitation Dataset (nClimGrid) Vose et al. (2014).

Table 1: **Predictive performance** comparisons over all datasets: we report classification accuracy for the classification tasks and MSE for the regression and forecasting tasks. Value in **bold** denotes the best result, and value with underline denotes the second-best result.

| | FL-60 (%) | CompCars (%) | TPT-48 ($10^{-3}$) | METR-LA ($10^{-3}$) | PEMS-BAY ($10^{-3}$) |
|---|---|---|---|---|---|
| FedAvg McMahan et al. (2017) | 97.9±9.7 | 88.2±1.3 | 2.6±0.9 | 112.4±43.2 | 33.5±38.5 |
| FedAvg-finetune | **100.0±0.0** | **90.0±5.8** | 2.6±1.0 | 20.0±7.8 | 13.2±12.9 |
| Per-FedAvg Fallah et al. (2020b) | 87.7±24.2 | 66.0±1.8 | 2.6±0.9 | 289.5±89.9 | 63.7±70.4 |
| pFedMe T Dinh et al. (2020) | **100.0±0.0** | 72.2±0.9 | 2.5±0.8 | 315.8±81.3 | 75.7±126.1 |
| SFL Chen et al. (2022) | 73.3±33.7 | 77.0±7.9 | 2.9±1.0 | 73.2±23.0 | 36.4±27.3 |
| pFedHN Shamsian et al. (2021) | **100.0±0.0** | 85.7±2.1 | **2.2±0.5** | 3.2±1.7 | 3.6±4.9 |
| *Panacea*-GN | **100.0±0.0** | 89.1±1.2 | 2.5±0.8 | 3.2±1.4 | 1.2±0.4 |
| *Panacea* | **100.0±0.0** | 88.2±1.2 | 2.6±0.7 | **1.0±0.2** | **0.4±0.4** |

Table 2: **Generalization capability** comparisons to novel clients: we report classification accuracy for the classification tasks and MSE for the regression and forecasting tasks. Value in **bold** denotes the best result, and value with underline denotes the second-best result.

| | FL-60 (%) | CompCars (%) | TPT-48 ($10^{-3}$) | METR-LA ($10^{-3}$) | PEMS-BAY ($10^{-3}$) |
|---|---|---|---|---|---|
| FedAvg McMahan et al. (2017) | 99.2±2.9 | 86.8±1.7 | 2.8±1.0 | 109.7±38.3 | 30.7±31.7 |
| FedAvg-finetune | **100.0±0.0** | 71.2±12.8 | 2.8±0.8 | 19.7±9.7 | 12.3±6.6 |
| Per-FedAvg Fallah et al. (2020b) | 80.8±22.3 | 66.4±3.1 | 2.8±1.0 | 274.8±80.1 | 59.4±64.2 |
| pFedMe T Dinh et al. (2020) | 42.5±37.5 | 72.0±2.2 | 3.5±1.5 | 305.3±83.2 | 98.5±97.5 |
| SFL Chen et al. (2022) | 62.5±37.0 | 77.5±7.0 | 3.3±1.2 | 70.8±24.9 | 33.7±19.3 |
| pFedHN Shamsian et al. (2021) | 82.5±32.8 | 86.4±3.8 | **2.5±0.7** | 3.2±1.5 | 3.0±2.7 |
| *Panacea*-GN | **100.0±0.0** | **87.9±1.9** | 2.6±0.9 | 3.1±1.2 | 1.2±0.5 |
| *Panacea* | **100.0±0.0** | 87.8±1.8 | 2.8±0.9 | **1.0±0.3** | **0.4±0.5** |

mean and standard deviation, are calculated over five independent runs. As for the data scale, FL-60 has 60 clients for the synthetic binary classification task where each client has 100 samples. TPT-48 has 48 clients and each client has 125 samples. In forecasting tasks, each sensor/detector is treated as a client; there are 325 clients in PEMS-BAY and 207 clients in METR-LA, respectively. More statistics of PEMS-BAY and METR-LA are listed in Appendix Table 3. For the image classification task, there are 30 clients (5 viewpoints × 6 YOMs) and 24,151 images in total. Moreover, all datasets in each client are randomly split into 80% / 20% for training / testing.

## 4.2 RESULTS AND DISCUSSION

**Averaged predictive performance across clients.** Table 1 reports the average accuracy or MSE across clients. In most cases, our approaches, including *Panacea* and *Panacea*-GN exhibit the best performance compared to other baselines. Among others, pFedHN achieves the second best or sometimes is on par with ours, demonstrating the optimality of hypernetwork-based approaches. The gap between our methods and pFedHN shows the validity of introducing graph information for balancing information sharing and uniqueness preservation among clients in the federation. In addition, we can observe that *Panacea* achieves significantly better performance than *Panacea*-GN for the forecasting tasks on two road network traffic datasets, indicating the necessity of incorporating the graph generator module. In contrast, *Panacea*-GN performs slightly better than *Panacea* for the regression task on TPT-48 and image classification on CompCars. It is interesting that performance gap between *Panacea* and *Panacea-GN* is related to the number of the clients. TPT-48 and CompCars have smaller number of clients (48 and 30) and show litte gap between *Panacea* and *Panacea-GN*. On the other hands, Panacea is significantly better than Panacea-GN on datasets with more clients (like from 60 to 325). We hypothesize that when the graph is small, our framework might overfit to the knowledge of the graph which can hurt the performance of the clients.

**Generalization to novel clients.** We also conduct experiments to evaluate the generalization capability of different federated learning algorithms. Specifically, we study an important learning setup where new clients join. In general, if models are shared across clients, new clients joining in would require fine-tuning the shared model, such as pFedME and Per-FedAvg. In contrast, it may not need fine-tuning or retraining in the federated learning framework based on hypernetworks, including pFedHN and our two methods *Panacea* and *Panacea*-GN. To verify the generalization capability of

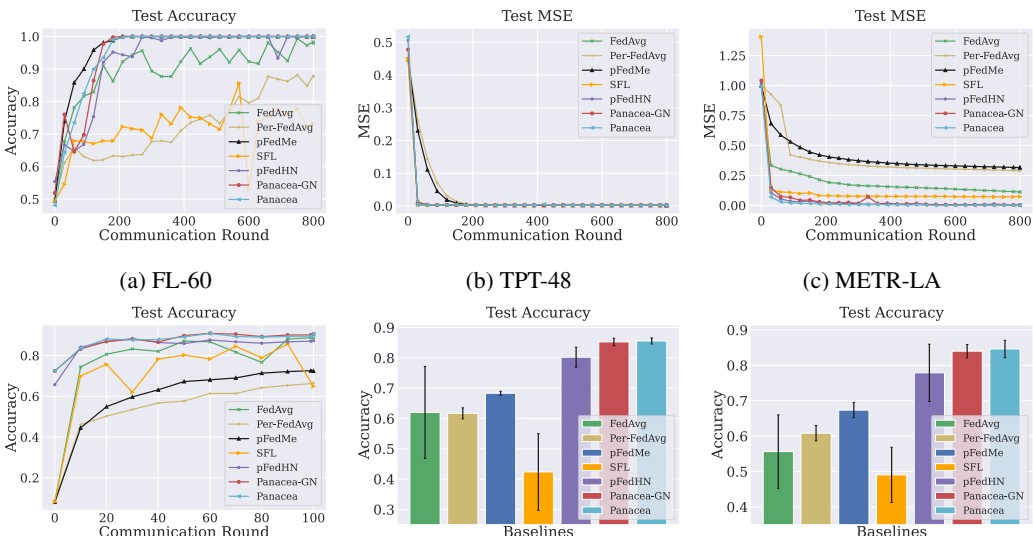

(a) FL-60           (b) TPT-48           (c) METR-LA

(d) CompCars     (e) CompCars (attack ratio: 30%)     (f) CompCars (attack ratio: 40%)

Figure 2: (a)-(e) compare convergence speed and final performance, and (f) compare final performance under various attack ratios on CompCars (More comparisons under different ratios are shown in Appendix E).

hypernetwork-based approaches, we take the client's initial embedding as the hypernetwork input to generate corresponding personalized models without fine-tuning and then evaluate the clients' averaged predictive performance with local test data. The results are shown in Table 2. We observe that our methods exhibit the best generalization capability in most cases. pFedHN also outperforms non-hypernework based methods like pFedMe, Per-FedAvg, and SFL. This can be explained by the fact that the federated hypernetwork framework essentially learns a meta-model over the distribution of clients with the proposed hypernetwork, thus having better generalization to novel clients.

**The visualization of convergence speed and task performance.** Figure 2 (a)-(d) shows the task performance during training with different personalized federated learning algorithms. We have the following observations: 1) Both SFL and Per-FedAvg show poor performance in the entire training process on FL-60; 2) generally, hypernetwork-based approaches, including pFedHN and ours, have the fastest convergence speed and exhibit comparable performance compared to other baselines; 3) both pFedMe and Per-FedAvg perform poorly on two real-world road network traffic datasets (PEMS-BAY shows a similar trend as METR-LA, shown in Appendix E); 4) *Panacea* shows a slightly faster convergence speed than *Panacea*-GN, reflecting the importance of training the graph hypernetwork with the graph generator.

**Robustness to malicious clients.** We conduct robustness evaluation over two classification tasks over FL and CompCars, respectively. Figure 2 (e)-(f) depicts the predictive performance of all approaches under 30% and 40% malicious clients. We can observe that our two methods — *Panacea* and *Panacea*-GN maintain superior performance even under a high attack ratio, verifying their resilience to malicious attacks.

## 5 CONCLUDING REMARKS

We introduced a new federated learning framework called *Panacea* for generating personalized models based on federated graph hypernetworks. The core of *Panacea* is the composition of a graph hypernetwork and a graph generator. *Panacea* has several advantages: it is agnostic to the target network architecture, can reinforce the collaboration among adjacent clients and generate personalized models across various application domains with better predictive performance, enjoys better generalization capability to unseen clients during training, is more resilient to malicious clients under label-flipping attacks. Extensive experiments on synthetic and real-world datasets have demonstrated the rationality and effectiveness of using *Panacea*, resulting in a new state-of-the-art for generating personalized models in federated learning settings.

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
