# OpenReview forum: "Graph-Relational Federated Learning: Enhanced Personalization and Robustness"
_ICLR.cc/2024/Conference — Submitted to ICLR 2024_

### Official Review · Reviewer_2W8m · 2023-10-31

**Soundness:** 3 good
**Presentation:** 2 fair
**Contribution:** 3 good
**Rating:** 5
**Confidence:** 3

**Summary:**

The paper proposes a novel personalized federated learning framework called Panacea, which is composed of a GNN encoder, an MLP, and a graph generator. In each communication round, the GNN encoder first refines client embeddings based on initial embedding and client relationship. Then, the MLP generates next round client parameters with refined embedding. Finally, the graph generator further preserves local preferences of each client through reconstruction. Empirical studies were performed to verify the effectiveness of the method.

**Strengths:**

1.	Compared with existing works, the author further Improves the personalization mechanism by considering client relationships in a graph manner and perform knowledge sharing over their embeddings.
2.	The framework can easily generalize to novel clients and is robust to malicious clients.

**Weaknesses:**

1.	It’s a bit confusing regarding where the client embedding comes from, especially for the real-world datasets. It seems that this work relies on human-designed client relationships and embeddings, which may be hard to realize in real-world scenarios.
2.	The assumption of the existence and accessibility of client graph is not very compatible with the experiment setting. As for unseen novel clients and malicious clients, it is typically unable to know their relationship with other normal clients (or only know incorrect relationship).  So how to exploit client relationship in these cases?
3.	The claim in the Lemma 3.1 part is not quite persuasive. From my perspective, the graph you adopt is not a weighted graph. Then if all the nodes are connected in the clique graph, the embedding of one node is just a plain average of all embeddings. How to fully recover the client embeddings just based on these average values? A more detailed explanation is needed here.

**Questions:**

1.	What is the purpose of the graph generator? Is it for the further improvement of latent client embedding or for improving given client relationships?

---

### Official Review · Reviewer_i3ns · 2023-11-08

**Soundness:** 2 fair
**Presentation:** 2 fair
**Contribution:** 3 good
**Rating:** 5
**Confidence:** 5

**Summary:**

This paper introduces Panacea, a hypernetwork-based federated learning (FL) framework that enables personalized model learning for individual clients. The key contribution of Panacea lies in its incorporation of the client relationship graph into hypernetwork learning, allowing for the consideration of heterogeneity in client data distribution. Through empirical studies across multiple tasks, the paper demonstrates the effectiveness of Panacea, showcasing its ability to generalize well to new clients and exhibit robustness.

**Strengths:**

1.	The paper focuses on an important research question, i.e., modeling personalized federated learning framework with the help of client relationship graph. By considering the correlations among clients, the system can capture the client similarities which promotes personalized client model learning.

2.	The paper is well organized and most of the clarifications provided are easy to follow and understand.

**Weaknesses:**

1.	The paper's technical novelty appears to be limited. In comparison to the existing work pFedHN, which employs an MLP as the hypernetwork, this paper merely replaces the MLP with a GNN module to capture client correlations. However, it does not introduce any specific techniques to address potential issues, such as the high computational cost associated with graph-based learning.
2.	The experiment utilizes manually constructed graphs, which may not naturally exist in the dataset. This introduces the possibility of inaccurate graphs and additional noise, potentially impacting the evaluation and diminishing support for the claim of effective personalization modeling through graph-based hypernetwork approaches.

**Questions:**

1.	Why the proposed method is robust to label-flipping attacks?
2.	What is the initial input of each client’s embedding into the GNN hypernetwork? Do different initialization methods have a significant impact on the model performance?
3.	Why do some baselines perform so poorly on the regression tasks? For example, the Per-FedAvg and pFedMe on the METR-LA and PEMS-BAY datasets.

---

### Official Review · Reviewer_vqGr · 2023-11-08

**Soundness:** 2 fair
**Presentation:** 3 good
**Contribution:** 2 fair
**Rating:** 5
**Confidence:** 5

**Summary:**

The paper proposes a graph-guided federated learning framework by incorporating a graph hypernetwork.

**Strengths:**

1. The targeting problem is an advanced setting in FL.

2. The proposed method is technique sounds.

3. The paper is well-written.

**Weaknesses:**

1. The novelty is limited. Using Hypernetwok in FL is an existing idea, and graph-guided FL is also an existing idea. This paper is the integration of these two ideas.

2.  Most hype-network-based methods highly rely on hyperparameter tuning that is difficult to be applied to real applications.

3. The method should be evaluated by FL benchmark datasets in experiments.

**Questions:**

1. Please refer to the weakness.

2. How the graph hypernetwork works is unclear. A more detailed description is required.

---

### Meta-Review · Area_Chair_CLw5 · 2023-12-09

**Metareview:**

The reviewers all complained about the limited novelty of this work compared to existing papers. The authors did not respond to the reviews. We recommend rejection.

**Justification For Why Not Higher Score:**

The novelty is limited

**Justification For Why Not Lower Score:**

NA

---

### Decision · Program_Chairs · 2024-01-16

Reject